# Novel Nanoparticles Based on *N*,*O*-Carboxymethyl Chitosan-Dopamine Amide Conjugate for Nose-to-Brain Delivery

**DOI:** 10.3390/pharmaceutics14010147

**Published:** 2022-01-08

**Authors:** Adriana Trapani, Stefania Cometa, Elvira De Giglio, Filomena Corbo, Roberta Cassano, Maria Luisa Di Gioia, Sonia Trombino, Md Niamat Hossain, Sante Di Gioia, Giuseppe Trapani, Massimo Conese

**Affiliations:** 1Department of Pharmacy-Drug Sciences, University of Bari “Aldo Moro”, 70125 Bari, Italy; filomena.corbo@uniba.it (F.C.); giuseppe.trapani@uniba.it (G.T.); 2Jaber Innovation s.r.l., 00144 Rome, Italy; stefania.cometa@jaber.it; 3Chemistry Department, University of Bari “Aldo Moro”, Via Orabona, 4, 70125 Bari, Italy; elvira.degiglio@uniba.it; 4Department of Pharmacy, Health and Nutritional Sciences, University of Calabria, Arcavacata di Rende, 87036 Cosenza, Italy; roberta.cassano@unical.it (R.C.); ml.digioia@unical.it (M.L.D.G.); sonia.trombino@unical.it (S.T.); 5Department of Medical and Surgical Sciences, University of Foggia, 71122 Foggia, Italy; mdniamat.hossain@unifg.it (M.N.H.); sante.digioia@unifg.it (S.D.G.); massimo.conese@unifg.it (M.C.)

**Keywords:** dopamine, intranasal administration, polymeric conjugates, polymeric nanoparticles, fluorescent microscopy

## Abstract

A widely investigated approach to bypass the blood brain barrier is represented by the intranasal delivery of therapeutic agents exploiting the olfactory or trigeminal connections nose-brain. As for Parkinson’s disease (PD), characterized by dopaminergic midbrain neurons degeneration, currently there is no disease modifying therapy. Although several bio-nanomaterials have been evaluated for encapsulation of neurotransmitter dopamine (DA) or dopaminergic drugs in order to restore the DA content in parkinsonian patients, the premature leakage of the therapeutic agent limits this approach. To tackle this drawback, we undertook a study where the active was linked to the polymeric backbone by a covalent bond. Thus, novel nanoparticles (NPs) based on *N*,*O*-Carboxymethylchitosan-DA amide conjugate (*N*,*O*-CMCS-DA) were prepared by the nanoprecipitation method and characterized from a technological view point, cytotoxicity and uptake by Olfactory Ensheating Cells (OECs). Thermogravimetric analysis showed high chemical stability of *N*,*O*-CMCS-DA NPs and X-ray photoelectron spectroscopy evidenced the presence of amide linkages on the NPs surface. MTT test indicated their cytocompatibility with OECs, while cytofluorimetry and fluorescent microscopy revealed the internalization of labelled *N*,*O*-CMCS-DA NPs by OECs, that was increased by the presence of mucin. Altogether, these findings seem promising for further development of *N*,*O*-CMCS-DA NPs for nose-to-brain delivery application in PD.

## 1. Introduction

Nowadays, notable challenges are emerging in the development of useful therapeutic approaches for the treatment of neurodegenerative diseases (NDs) [1]. Thus, it is commonly accepted that the main obstacle for drug delivery in therapeutic concentrations to the brain is constituted by the blood brain barrier (BBB), which maintains the brain homeostasis and prevents the entry of substances that can cause neuronal damage [2]. This obstructing effect of the BBB is due to brain capillary endothelial cells, which limit the transcellular transport, as well as to the tight junctions between cells, that restrict the paracellular transport [3]. In general, in the case of NDs the BBB, even though compromised and more permeable, is still capable to protect the brain from the entry into it of dangerous molecules [2]. Hence, to treat NDs with the available drug delivery systems (DDS) and to slow down their progression, it is necessary to use high doses of therapeutic agents to gain appropriate BBB crossing [4]. In addition, the delivery of the therapeutic agent is often made by prolonged infusion or highly invasive procedures. Due to all these unfavourable factors, poor results have been observed in clinical trial studies employing current DDS for the treatment of neurological diseases [1].

A widely investigated approach to bypass the BBB is represented by the intranasal delivery of therapeutic agents allowing the direct access via nose-to-brain of the active using mainly the olfactory and trigeminal nerve transport pathways [5,6,7,8]. Moreover, intranasal delivery may also allow the indirect drug absorption by the lymphatic system and its successive transport into the systemic circulation [6]. In some circumstances, the nasal drug administration route may be even an effective alternative to the oral one [9]. This last, although characterized by patient compliance and the endorsement of pharmaceutical industry [10,11], may have some disadvantages for poorly soluble and/or unstable drug molecules in the gastrointestinal fluids [11]. However, it has been also demonstrated that nose-to-brain delivery with several molecules is unfeasible since they do not possess suitable properties, as high molecular weight, limited diffusion through mucus barrier, binding to mucin and mucociliary clearance (which reduce the residence time in the nose) to reach the brain in therapeutic amount. Other factors accounting for reduced brain bioavailability are related to the limited volumes that can be intranasally administered, and/or to enzymatic degradation in the olfactory mucosa. Hence, it is recognized that nose-to-brain delivery is practically limited only to potent drug molecules, typically those with effective plasma concentrations in the ng/mL (or lower) range [6].

In addition, nowadays most promising strategies for the treatment of NDs are also those based on drug encapsulation in nanocarriers which, appropriately designed, are able to improve the BBB crossing, overcoming so the problems showed by the current DDS employed for neurological diseases treatment [12,13,14]. This favourable effect of using nanocarriers is partially explained by the protection of the drug from degradation and/or efflux back into the nasal cavity [12] or because these nanoformulations can efficiently overcome the BBB, exploiting the normal physiological mechanisms of transport [13] including those involved in the intranasal route [15].

In the developed Countries, Parkinson’s disease (PD) is the second most common neurodegenerative diseases after Alzheimer’s disease [16] and it is characterized by dopaminergic midbrain neurons degeneration. Consequently, the patient presents the classical motor impairments symptoms of PD including tremor, rigidity, bradykinesia and postural instability. As the disease progresses, many PD patients show not only motor symptoms but also non-motor ones such as sleep disorders or depression, with a great impact on their quality of life. At present, there is no disease modifying therapy for PD, similarly for other NDs [17]. Current treatments of PD are based on the so-called “Dopamine replacement strategies (DRS)” aimed at modulating the dopamine (DA) levels administering DA agonists and this is known as “DRS by pharmacological approaches” [18]. On the other side, the DRS concept to restore neuron functionality into the brain has been demonstrated in preclinical studies harnessing exogenous/endogenous neural stems or neurotrophic factors and it is known as “cell-based (or -replacement) therapy” [18,19].

In the last decade several natural and synthetic polymers have been evaluated for nanoencapsulation of free DA or dopaminergic drugs in the context of “DRS by pharmacological approaches” [20,21,22,23,24,25,26]. However, a possible drawback of this strategy may be the premature leakage of the encapsulated therapeutic agent which could further reduce the low brain bioavailability above mentioned consequent to the BBB occurrence. In order to limit this problem, we decided to evaluate the nanoencapsulation of therapeutic agent-polymer conjugates where the leakage of the active should be limited or missing at all being linked to the polymeric backbone by a covalent bond. In order to offer several advantages including enhanced drug solubilisation, prolonged circulation and improved mucoadhesive properties, polymer-drug conjugates can be nanostructured leading to polymeric nanoparticles which may be able to cross the BBB by endocytosis [27,28]. Our interest in using chitosan and its derivatives as biopolymers for DRS by pharmacological approaches [21,22] led us to evaluate *N*,*O*-Carboxymethyl chitosan-DA conjugates where the polymeric chain is linked to the neurotransmitter moiety by a cleavable ester or amide bond [29,30].

The aim of the present work was to prepare novel nanoparticles (NPs) based on *N*,*O*-Carboxymethyl chitosan-DA amide conjugate (*N*,*O*-CMCS-DA) [30] and to evaluate their potential for nose-to-brain delivery of DA to be administered as such or formulated by using an appropriate biomaterial [1,18]. It is now recognized, indeed, that the use of biomaterials to deliver drugs for the treatment of neurological disorders, as PD, is favourable because they can allow appropriate release kinetics or prevents degradation of the therapeutic agent [1,18]. On the other hand, they may constitute a physical scaffold supporting cellular survival, integration and differentiation in cell-based therapies [18]. Possible biomaterials for neurological disorders could be mucoadhesive chitosan-based hydrogels which could constitute a favourable matrix in contact with olfactory epithelium and they have been extensively used in biomedical applications [31,32,33]. Indeed, the use of appropriate mucoadhesive systems, as such hydrogels, reduces the mucociliary clearance and increases the retention time in the nasal mucosa [15].

This work is in the context of our ongoing research project on carbohydrate based nanocarriers for smart therapeutic applications, such as overcoming of the BBB and treatment of neurological disorders. Carbohydrates, as chitosan and its derivatives, indeed, have received considerable attention for brain delivery due to their natural origin, inherent biodegradability and biocompatibility. The feasibility of using carbohydrate-based DDS in brain delivery has been demonstrated both with oligomeric as well as polymeric carbohydrates [30,34]. For the purpose mentioned, *N*,*O*-CMCS-DA NPs were successfully prepared by nanoprecipitation method starting from the *N*,*O*-CMCS-DA amide conjugate [30]. These nanocarriers were characterized from a technological viewpoint including their internal structure, thermal stability, surface features, release profile and physical stability. Cytotoxicity evaluation of *N*,*O*-CMCS-DA NPs were carried out by MTT assays on Olfactory Ensheathing Cells (OECs). Flow cytometry and epifluorescence microscopy were used to evaluate internalization of labelled *N*,*O*-CMCS-DA NPs by OECs.

## 2. Materials and Methods

### 2.1. Materials

*N*,*O*-Carboxymethyl Chitosan (*N*,*O*-CMCS, Molecular weight in the range of 30–500 kDa as stated by the supplier, deacetylation degree, 94.2%; viscosity 22 mPa**•**sec) was purchased from Heppe Medical Chitosan GmbH (Halle, Germany). Dopamine hydrochloride (DA**•**HCl), porcine stomach mucin (type II, bound sialic acid ~1%) Polyvinylalcohol (PVA, polymerization degree 500), Fluorescein 5(6)-isothiocyanate (FITC) and PBS were supplied by Sigma-Aldrich (Milan, Italy). Dialysis tubes with a MWCO-3500 kDa were purchased from Spectra Labs (Rome, Italy). Throughout this work, double distilled water was used. All other chemicals used were of reagent grade. Dopamine-fluorescein isothiocyanate (FITC-DA) was synthesized starting from DA and FITC as described by Carta et al. [35].

### 2.2. Quantitative Determination of Dopamine and N,O-Carboxymethylchitosan-DA Amide Conjugate (N,O-CMCS-DA)

The quantitative determination of DA was carried out by HPLC as previously reported [30] using 0.02 M potassium phosphate buffer, pH 2.8: CH_3_OH 70:30 (*v*:*v*) as mobile phase and the elution of the column in isocratic mode took place at the flow rate of 0.7 mL/min. Under such chromatographic conditions, the retention times of DA and *N*,*O*-CMCS-DA were 5.5 min and 4.9 min, respectively [30].

### 2.3. Preparation of N,O-CMCS- or N,O-CMCS-DA-Loaded NPs (N,O-CMCS- or N,O-CMCS-DA-NPs) and Corresponding FITC Labelled NPs (FITC-N,O-CMCS or FITC-N,O-CMCS-DA NPs)

*N*,*O*-CMCS or *N*,*O*-CMCS-DA NPs were prepared starting from the commercially available *N*,*O*-CMCS or from *N*,*O*-CMCS-DA prepared as previously reported [30], respectively, according to nanoprecipitation method previously reported by Musumeci et al. with some modifications [36]. A dispersion of *N*,*O*-CMCS or *N*,*O*-CMCS-DA at the concentration of 10 mg/mL in acetone was sonicated in a sonicator water bath (Branson Sonifiers, Rungis, France) for 40 min at room temperature (r.t.). Afterwards, 50 μL of the resulting dispersion were poured at r.t. into 0.5 mL of an aqueous PVA solution (1 mg/mL) and the resulting mixture was quickly sonicated by probe sonication with a Vibra-cell sonicator (Sonics and Materials, Market Harborough, UK) for 8 min in an ice water bath. Then, the mixture was centrifuged (Eppendorf 5415D, Hamburg, Germany) at 13,200 rpm for 45 min. The resulting pellet was employed for following studies and the consequent supernatant was discarded. Lyophilized samples of *N*,*O*-CMCS-DA NPs were obtained using a Lio Pascal 5P (Milan, Italy) apparatus.

To achieve the corresponding FITC fluorescently labelled NPs, i.e., FITC-*N*,*O*-CMCS or FTIC-*N*,*O*-CMCS-DA NPs, the same procedure was adopted starting from FITC-*N*,*O*-CMCS [29] or FITC-*N*,*O*-CMCS-DA [30], respectively. The labelling efficiency of the resulting fluorescent NPs was determined as previously reported [37].

### 2.4. Physicochemical Characterization of Nanoparticles Prepared

Particle size and polydispersity index (PDI) of NPs were measured at 25 °C after dilution in double distilled water (1:1, *v*:*v*) by using a Zetasizer NanZS (ZEN 3600, Malvern, UK) apparatus according to photon correlation spectroscopy (PCS) mode. The determination of the zeta-potential was also performed at 25 °C using laser Doppler anemometry (Zetasizer NanoZS, ZEN 3600, Malvern, UK) after dilution 1:20 (*v*:*v*) in the presence of KCl (1 mM, pH 7). The particle size, PDI and zeta potential values were each measured in triplicate and the results are shown as the mean ± SD.

The morphology of NPs was acquired by cryogenic transmission electron microscopy (Cryo-TEM) using a Hitachi 7700 electron microscope operating at a temperature of 105 K and an acceleration voltage of 100 KV. The procedure for deposition of the samples has been previously described [38] and it involved the deposition of a drop of NP suspension on copper grids covered with an amorphous carbon film. After removing the excess solution, the sample was vitrified by immersion in liquid ethane maintained just above its freezing point. Then, the sample was transferred to the Gatan 626 cryo holder. The sample was protected against atmospheric conditions during the entire procedure to prevent the formation of ice crystals. The digital images were acquired with an AMT-XR-81 camera and processed with the EMIP software, version 3DFSC.

To determine the Encapsulation Efficiency of neurotransmitter (i.e., E.E. DA%) or FITC (i.e., E.E. FITC%) in the NPs, after separation from the resulting pellet (Section 2.3), the supernatant was analyzed by HPLC for DA or *N*,*O*-CMCS-DA content, according to the quantitative determination above described, whereas the FITC content was determined by a fluorimetric method as previously reported [30].

The encapsulation efficiency (E.E.%) was calculated as follows:E.E.% = *w_i_* − *w*_s_/*w_i_* ∗ 100
where *w_i_* is total initial weight of DA or *N*,*O*-CMCS-DA or FITC and *w*_s_ is the weight of DA or *N*,*O*-CMCS-DA or FITC determined in the supernatant.

### 2.5. Solid State Studies

#### 2.5.1. Fourier Transform Infrared (FT-IR) Spectroscopy

FT-IR spectra were obtained in KBr discs using 2–5 mg of pure DA•HCl, *N*,*O*-CMCS, *N*,*O*-CMCS-DA, and lyophilized *N*,*O*-CMCS-DA NPs using a Perkin Elmer 1600 FT-IR spectrometer (Perkin Elmer, Milan, Italy). The analysis was carried out at r.t. in the range of 4000–400 cm^−1^ at a resolution of 1 cm^−1^ [33].

#### 2.5.2. Differential Scanning Calorimetry (DSC)

DSC analyses on pure DA•HCl, *N*,*O*-CMCS, *N*,*O*-CMCS-DA and lyophilized) *N*,*O*-CMCS-DA NPs were carried out using a Mettler Toledo DSC 822e instrument. The apparatus was calibrated with indium and in each run the heating rate of 5 °C/min was used in the range of 25–275 °C. About 5 mg of each sample filled the standard aluminum sample pans for analysis and an empty pan was used as reference. Analyses were performed under nitrogen flow of 20 cm^3^/min. Each experiment was carried out in triplicate [39].

#### 2.5.3. Thermogravimetric Analysis (TGA)

Thermogravimetric Analysis (TGA) has been performed on a PerkinElmer TGA-400 instrument (PerkinElmer Inc., Waltham, MA, USA), heating samples (5–10 mg) from 30 to 600 °C in nitrogen-saturated atmosphere. The flow rate was set to 20 °C/min. For each specimen, thermograms (TG) and derivative (DTG) curves have been acquired and analyzed with the TGA Pyris software, Version 13.3.1.0014.

#### 2.5.4. X-ray Photoelectron Spectroscopy (XPS)

X-ray photoelectron spectroscopy (XPS) analysis has been carried out to gain information on the surface chemical composition of the *N*,*O*-CMCS-DA NPs. The analysis was performed using a scanning microprobe PHI 5000 VersaProbe II (Physical Electronics, Chanhassen, MN, USA), equipped with a monochromatized AlKα X-ray radiation source. The specimens were analyzed in HP mode (scanned size: 1400 × 200 µm, X-ray take-off angle of 45°). For each sample, survey scans (pass energy 117.4 eV) and high-resolution spectra (pass energy 29.35 eV) were acquired in FAT mode. The interpretation of the results by means of a curve-fitting procedure was carried out using MultiPak software package (version 9.6.0.8). The peak area were normalized by correction with empirically derived sensitivity factors, according to MultiPak library, thus enabling comparison of data from different elements. Charge referencing was performed by setting the lower binding energy C1s photo-peak at 284.8 eV (i.e., C1s hydrocarbon peak).

### 2.6. Physical Stability of N,O-CMCS-DA NPs on Storage

The physical stability of *N*,*O*-CMCS-DA NPs was evaluated measuring their particle size after incubation upon storage at 4 °C up to 3 months as well as at 25 °C over one week [28] and at 37 °C up to 24 h. The particle size was measured at different time intervals according to the description reported in Section 2.4.

### 2.7. In Vitro Release in Simulated Nasal Fluid/Mucin

A weighted amount of *N*,*O*-CMCS-DA NPs corresponding to 1–1.2 mg of DA was dispersed in 20 mL of Simulated Nasal Fluid (SNF) [29] mixed with 0.25% (*w*/*v*) of mucin (pH of the mixture = 6). Prior to start release test, the medium was thermostated at 37 ± 0.1 °C in an agitated (40 rpm/min) water bath (Julabo, Milan, Italy) and the release study was conducted for 55 h. At scheduled time points, 0.8 mL of the receiving medium were withdrawn and replaced with 0.8 mL of fresh medium. Then, each sample withdrawn was centrifuged at 16,000× *g* for 45 min, (Eppendorf 5415D, Germany), and the amounts of the neurotransmitter delivered were determined in the resulting supernatants following the analytical protocol described in Section 2.2, and plotted against the time. All the release experiments in SNF containing mucin were performed in triplicate.

### 2.8. Cytotoxicity Studies with Olfactory Ensheathing Cells (OECs)

OECs were obtained from olfactory bulbs of mouse P2 as previously reported [40,41].

Cells were grown in Dulbecco’s Modified Eagle’s medium (DMEM) supplemented with 10% FBS and bovine pituitary extract, with regular media changes twice a week. Subsequently, cells (3 × 10^4^ cells/well) were seeded in 96-well plates and exposed to *N*,*O*-CMCS-DA NPs corresponding to DA concentrations of 0.3, 1.17, 4.7, 18.75 and 75 μM. Twenty-four h after treatments, cell viability was evaluated by MTT (3-(4,5-dimethylthiazol-2-yl)-2,5 diphenyl tetrazolium bromide), as previously described [30]. The cell viability was calculated as follows:% viability = [(Optical density {OD} of treated cell − OD of blank)/(OD of vehicle control − OD of blank) × 100], 
considering untreated cells as 100%. Cells treated with 1% Triton X-100 were used as positive control.

### 2.9. Flow Cytometry

OECs (plated at the number of 50,000 per each well of a 24-well plate) were incubated with FITC-*N*,*O*-CMCS-DA NPs in order to obtain the final concentrations of 18.75 and 75 µM DA in the presence or absence of 2.5% mucin. After 2 h, each well was treated with 0.04% trypan blue in PBS (in order to quench extracellular fluorescence), trypsinized, resuspended in 0.5 mL of PBS, and cells were evaluated by the Amnis Flowsight IS100 (Merck). Brightfield scatter plots obtained by plotting Area on *x*-axis vs Aspect Ratio on *y*-axis were generated, then single cells events were gated, and finally 10,000 single-cell events for sample were acquired. The percentage of green positive cells (channel 2, 488 nm excitation laser) and mean fluorescence were analyzed using Amnis IDEAS software, Version 6.0 [42].

### 2.10. Epifluorescence Microscopy

OECs were seeded onto glass coverslips in 24-well plates (5 × 10^4^ cells/well). After 24 h, the medium was removed, cells were washed with PBS and then exposed to two different FITC*-N*,*O*-CMCS-DA NPs concentrations (corresponding to 18.75 and 75 µM DA), or FITC-DA (18.75 and 75 μM) [35], in the presence or absence of 2.5% mucin for 2 h. After incubation, cells were washed with PBS, then fixed with a solution containing 1% sucrose and 1% paraformaldehyde in PBS, for 5 min. Then, coverslips were mounted and nuclei were counterstained with Vectashield Antifade Mounting Medium with 40,6-diamidino-2-phenylindole (DAPI) (Vector Laboratories Inc., Burlingame, CA, USA). The preparations were viewed under an epifluorescence microscope (Nikon Eclipse Ni) using a 60× magnification. Negative controls for background fluorescence were OECs incubated with medium only for 2 h.

### 2.11. Statistics

Statistical analyses were carried out by Prism v. 5, GraphPad Prism 5.0. Data were expressed as mean ± SD. Multiple comparisons were based on one-way analysis of variance (ANOVA) with the either Bonferroni’s or Tukey’s post hoc test and differences were considered significant when *p* < 0.05.

## 3. Results

### 3.1. Formulation and Characterization of DA/CSNPs

The nanoprecipitation method previously reported [36] allowed us to prepare *N*,*O*-CMCS or *N*,*O*-CMCS-DA NPs by adding a sonicated dispersion of *N*,*O*-CMCS or *N*,*O*-CMCS-DA in acetone to an aqueous PVA solution (1 mg/mL). The resulting mixture was sonicated and centrifugated providing the desired nanocarriers. Similarly, the corresponding FITC fluorescently labelled NPs, i.e., FITC-*N*,*O*-CMCS or FITC-*N*,*O*-CMCS-DA NPs were prepared. Table 1 summarizes the main physicochemical features of the different NPs prepared. As can be seen, NPs arising from *N*,*O*-CMCS conjugated to small molecules (DA or FITC) were lower in size than the unconjugated ones (i.e., *N*,*O*-CMCS NPs) which resulted the biggest (252 ± 33 nm/289 ± 50 nm and 608 ± 58 nm, respectively). When both the small molecules were conjugated to the same polymer backbone ((i.e., *N*,*O*-CMCS) an intermediate mean diameter value was observed (425 ± 28 nm). In all cases examined a broad size distribution of the particle population was observed as proved by the PDI values ranging between 0.34 and 0.62. A bimodal size distribution was evidenced by PCS for *N*,*O*-CMCS-DA NPs (Figure 1a).

Zeta potential measurements were all negative values due to the corresponding negative charges on the surface of *N*,*O*-CMCS backbone. Interestingly, all NPs obtained from *N*,*O*-CMCS conjugated to DA and/or FITC showed zeta potential values significantly different from the *N*,*O*-CMCS NPs, the most negative being *N*,*O*-CMCS-DA NPs (i.e., −32.4 ± 1.6 mV). The E.E. DA% for *N*,*O*-CMCS-DA NPs was essentially quantitative and only a slight reduction occurred for DA content referring to FITC-*N*,*O*-CMCS-DA NPs. When fluorescent NPs were achieved by covalent linkage of FITC to the *N*,*O*-CMCS polymer, the E.E. FITC% resulted quantitatively bound to the final NPs either in the presence or in the absence of neurotransmitter conjugated.

As for the particle morphology of *N*,*O*-CMCS-DA NPs, cryo-TEM visualizations showed spherically shaped particles which appear stable under electronic beam used for TEM visualization, namely no collapsing of the colloids was denoted during beam irradiation (Figure 1b). Moreover, it was deduced that no aggregated particles there were as well as a thick layer was observed around the particles of *N*,*O*-CMCS-DA NPs resulting in a core-corona structure. By comparing the particle size results deduced from TEM with those measured by PCS, it is clear that using the former approach, a smaller mean diameter value for *N*,*O*-CMCS-DA NPs can be obtained. Such difference should be attributed to the fact that PCS measurements were carried out in suspension providing hydrodynamic diameters, while TEM analyses were performed at the dried state [43].

### 3.2. Solid State Studies

To increase our knowledge on the solid state features of *N*,*O*-CMCS-DA NPs herein prepared, spectrophotometric (FT-IR) and thermal analysis (DSC and TGA) studies were performed and the corresponding results are shown in Figure 2 and Figure 3. Moreover, details on the surface composition of the *N*,*O*-CMCS-DA NPs were obtained by XPS analysis. The absorption bands of pure DA•HCl (Figure 2) in the range 1600–1650 cm ^−1^ were not observed in FT-IR spectrum of *N*,*O*-CMCS-DA NPs, similarly to that previously observed for *N*,*O*-CMCS-DA [30]. The broad peak at 1622 cm^−1^ in the FT-IR spectrum of *N*,*O*-CMCS as well as the missing absorption at 1730 cm^−1^ suggest that the starting carboxymethyl chitosan used to prepare *N*,*O*-CMCS-DA [30] is in the –COONa form and, hence, the commercial polymer is a polyanionic macromolecule [44,45]. Such broad absorption is still present both in the *N*,*O*-CMCS-DA and *N*,*O*-CMCS-DA NPs, even though less intense (1640 cm^−1^ and 1638 cm^−1^, respectively). However, the latter nanocarriers, besides the intense peak at 3416 cm^−1^ due to the –OH absorption band, showed also the occurrence of a clear absorption peak at 1731 cm^−1^ which is absent in the starting *N*,*O*-CMCS-DA and, hence, it should be a distinctive feature of these nanocarriers.

The DSC thermogram of pure DA•HCl (Figure 2) showed an intense endothermic peak at 253 °C corresponding to the melting of the drug [46]. In the thermograms of *N*,*O*-CMCS and *N*,*O*-CMCS-DA no distinct peaks were present suggesting an amorphous internal structure for these macromolecules (Figure 2). As previously reported [47], for the polysaccharide *N*,*O*-CMCS the endothermal broad peak at 142 °C should be due to the water loss. *N*,*O*-CMCS-DA alone showed a low shift of the peak corresponding to the melting point of DA hydrochloride (i.e., 248 °C) while *N*,*O*-CMCS-DA NPs exhibited a distinct endothermic peak at high temperature (i.e., 262 °C), presumably due to the particle thermal decomposition.

TGA thermograms of selected analytes were shown in Figure 3. As far as pure DA•HCl is concerned, the main pyrolytic event occurred at a Tpeak = 327 °C, evidencing a high drug stability under non-oxidative conditions, with a residue at 600 °C of 29 %. In the case of pure *N*,*O*-CMCS, a first event was associated to the water/volatile loss (Tpeak = 56.3 °C, ΔW = 10.6%); then, two partially overlapped events occurred (295 °C and 307.4 °C, ΔW = 17.5 and 8.9%, respectively). The second event could be probably associated to the removal of carboxylate from polysaccharide, with carbon dioxide release from samples; the third degradation was due to the possible NH_2_ detachment in the form of ammonia (NH_3_) released. The residue at 600 °C was equal to 51%. Finally, *N*,*O*-CMCS-DA NPs showed a low water/volatiles loss (Tpeak = 65.8 °C, ΔW = 0.2%) and two partially overlapped events in a large peak. (298.6 and 320.5 °C, with ΔW = 28.7 and 44.1%, respectively).

XPS analysis was carried out on *N*,*O*-CMCS-DA NPs, as well as on their precursors (i.e., pure CMCS and pure DA•HCl). In Table 2, the atomic percentages detected on the analyzed samples are reported.

In particular, the N1s signal was acquired and an accurate curve-fitting was performed in order to supply evidence of the amide linkage presence on the NPs surface. N1s of pure DA consisted prevalently of protonated amine moieties, since DA was in the hydrochloride form, as already reported [48]. As far as the N1s signal of CMCS protection (Figure 4a), the amine groups of chitosan (falling at 399.4 eV) were partially transformed in amide groups (falling at 401.3 eV), due to the carboxymethylation process. The percentage of amide groups was about 22% of the total nitrogen signal.

As far as the *N*,*O*-CMCS-DA NPs N1s signal is concerned (Figure 4b), a significantly higher contribution of the amide groups (falling at 401.0 eV) was detected, with a percentage of amide groups that increased up to 40%.

### 3.3. Physical Stability of N,O-CMCS-DA NPs on Storage

*N*,*O*-CMCS-DA NPs were exposed at 4 °C, 25 °C and 37 °C, for different time intervals and their particle size was determined (Figure 5). Overall, NPs increased their mean diameters in a time dependent manner, and, precisely, particle size was doubled at the latest time points, both at 37 °C and 4 °C (Figure 5a–c). However, neither aggregates nor colour changes were seen under incubation, irrespectively of the tested temperature.

### 3.4. In Vitro Release in Simulated Nasal Fluid

Figure 5d shows the release profile obtained after incubation of *N*,*O*-CMCS-DA NPs in the medium SNF/mucin. It should be noted that from the retention times of the chromatographic peaks in the range between 4.9–5.5 min it was not possible to assign each of them unequivocally to DA containing substances (i.e., pure neurotransmitter or *N*,*O*-CMCS-DA conjugate oligomers) or to a mixture of both. Therefore, the cumulative DA released shown on *y*-axis of Figure 5d should be actually referred to DA containing substances or to their mixture. As shown, a prompt release of DA was observed, reaching 20% of the total DA delivery after 3 h and this amount resulted constant up to more than two days of release. During all this release time no change in color was detected, ascribable to DA degradation [48].

### 3.5. Cytotoxicity Studies in OECs

OECs are cells ensheating the olfactory nerve in its way to the olfactory bulb, thus it is worth to understand whether *N*,*O*-CMCS-DA NPs were cytotoxic to these cells. Cell viability was examined by the MTT test and was determined 24 h after incubation of cells with *N*,*O*-CMCS-DA NPs in order to obtain different DA concentrations. As shown in Figure 6, NPs were slightly toxic to OECs at all DA concentrations, with a maximal reduction of cell viability of around 10% with 75 μM DA.

### 3.6. Uptake Studies

OECs would transport DA through the olfactory nerve once DA carried by their NP cargo is taken up by these cells. To ascertain whether OECs are capable to internalise *N*,*O*-CMCS-DA NPs epifluorescence studies were carried out with FITC-NPs carrying DA or FITC-DA. The DA concentrations of 18.75 and 75 μM were chosen to deliver an appropriate therapeutic dose [38,49] while causing low-level cytotoxicity for OECs (Figure 7). Cytofluorimetric analysis of the uptake at 2 h showed that the DA concentration did not change the number of positive cells, which was around 2%, while the presence of 2.5% mucin increased significantly these percentages with both concentrations (Figure 7a). Another parameter that is possible to study by cytofluorimetry is the mean fluorescence intensity (MFI), an indication of distribution of NPs among cells. Interestingly, the higher the DA concentration the higher the MFI, although not significantly, whereas the MFI was not substantially changed by the presence of mucin (Figure 7b). Overall, these data indicate that mucin determined NP uptake by more cells.

To confirm cytofluorimetric data, we performed an epifluorescence study. Very few cells were associated with FITC-*N*,*O*-CMCS-DA NPs at 18.75 μM DA, evidenced by dots closed to nuclei, whereas more cells incubated with FITC-*N*,*O*-CMCS-DA NPs at DA concentration of 75 μM showed perinuclear dots or diffuse perinuclear staining (Figure 8a,b). In the presence of mucin, the number of cells that had taken up FITC-*N*,*O*-CMCS-DA NPs increased in comparison with the condition in the absence of mucin at the corresponding concentration (Figure 8c,d). Notably, with 75 μM and mucin, the dots enlarged and more than one dot was associated with single cells (Figure 8d). Uptake was not observed with FITC-*N*,*O*-CMCS-DA at concentration of DA equal to 18.75 μM (not shown) and 75 μM, either in the absence or presence of mucin (Figure 8e,f). Controls in the absence or presence of mucin were devoid of any signal that could confound the specific one (Figure 8g,h). Overall, although cytofluorimetry has a quantitative and more sensitive outcome, both assays showed that mucin determines a higher number of cells that had taken up FITC-*N*,*O*-CMCS-DA NPs.

## 4. Discussion

In this work, we describe the preparation and characterization of *N*,*O*-CMCS-DA NPs as potential nanostructured carriers to be intranasally administered as such or formulated in combination with mucoadhesive hydrogels [50] to restore neuron functionality into the brain of parkinsonian patients. The potential of *N*,*O*-CMCS-DA amide conjugate to treat PD has been previously described together with its release profile [30]. For this purpose, we followed the nanoprecipitation method [36] to obtain not only the required NPs but also to produce *N*,*O*-CMCS NPs and the corresponding labelled NPs, i.e., FITC-*N*,*O*-CMCS or FTIC-*N*,*O*-CMCS-DA NPs as well. However, as shown in Table 1, the nanocarriers prepared were characterized by mean diameters >200 nm as well as by a broad size distribution as demonstrated by the high PDI values and PCS measurements. Similar results have been already observed in the literature when nanoprecipitation method has been selected to prepare polymeric NPs [51]. To reduce the unfavourable physicochemical NPs features, it has been emphasized the importance to control some crucial factors during the various steps of the nanoprecipitation method [52]. As well known, this last involves the use of two miscible phases, namely an organic phase (the solvent) in which the polymer and the active are dissolved and an aqueous phase (the non-solvent) [51,52]. Thus, the addition of solvent to non-solvent leads to a decrease of polymer solubility (i.e., supersaturation step) and then formation of primary particle nuclei (i.e., nucleation step) followed by their size growth (i.e., growth step) up to particle precipitation takes place (i.e., coagulation step) [52]. According to the literature, an accurate control of the factors involved in the steps of this preparative method may allow the production of NPs with satisfactory physicochemical features and better than those obtained by other methods (e.g., emulsification-solvent evaporation method) [52]. One of these critical factors is the mixing rate of phases. Indeed, poor mixing conditions provides low nucleation rate with the growth of few particles leading to few big NPs formation, while appropriate mixing conditions induces high nucleation rate with a larger population of small particles. The uniform growth of the particle nuclei is another important factor which plays a key role in determining a mono- or pluri-disperse particle population and, hence, their PDI value [51]. In this context, it is noteworthy the approach used by Craparo and coworkers [51] to remove the excess of surfactant (i.e., PVA) used in their preparation of Rhodamine B loaded PLGA–PEG NPs prepared by nanoprecipitation method. Such purification method allowed the authors to reduce both the particle size and improve the colloidal stability of these nanocarriers. Altogether, all these outcomes suggest that a notable improvement of the physicochemical properties of the resulting particles herein studied should result optimizing both the reaction conditions and removing the excess of surfactant.

In addition to the intrinsic factors of the preparative method, in the case herein studied there are further specific issues which need to be considered to account for the mean diameters (>200 nm) of *N*,*O*-CMCS-DA NPs and their PDI values (in the range 0.34–0.62) in agreement with a bimodal size distribution evidenced by PCS. In particular, we refer to the wide range of molecular weight (30–500 kDa) of the starting commercially available polymer used (*N*,*O*-CMCS). Such very high polydispersion in molecular weight of the starting polymer not only brings about a lower solubility in acetone selected as organic solvent but also affects all the successive steps above summarized for the production of NPs by nanoprecipitation. As a consequence, we did not observe a true polymer solution in acetone but a dispersion and, hence, the need of using also mixture sonication before the addition of the aqueous surfactant (PVA) solution. Moreover, under such conditions, even the uniform growth of the nuclei may be more difficult enhancing, so, the possibility to produce a pluri-modal particle population.

Another interesting result reported in Table 1 is that NPs arising from *N*,*O*-CMCS conjugated to small molecules (DA or FITC) were very lower in size than the unconjugated ones (i.e., *N*,*O*-CMCS NPs), while, when both DA or FITC were conjugated to the same polymer, an intermediate size was observed. At present, the reason accounting for this behaviour is not clear, but our hypothesis is that the conjugation with such small molecules brings about a conformational reorganization of the polymer leading to NPs shrinkage. In particular, we think that specific π-π interactions between the aromatic moieties in *N*,*O*-CMCS-DA NPs or in FITC*-N*,*O*-CMCS NPs may cause the shrinkage of the corresponding NPs. This effect is less intense when DA and FITC were both grafted onto the same polymer backbone as occurs for FITC*-N*,*O*-CMCS-DA NPs.

From the zeta potential values some noteworthy insights can also be deduced (Table 1). Thus, the lowest negative zeta potential occurs in the case of *N*,*O*-CMCS NPs (i.e., −9.2 ± 0.7 mV) which resulted also the biggest in size particles. It suggests that in such NPs the polymeric chains are looser and the negative charge density on the surface of these particles is lower. In the case of *N*,*O*-CMCS-DA NPs, it can be deduced that the corresponding polymeric chains are more dense and higher is the negative charge density on their surface. Moreover, the high enough zeta potential observed (−32.4 ± 1.6 mV) suggests that these particles should be endowed with sufficient colloidal stability.

As for the results of E.E % (Table 1), the high percentages observed both for DA and FITC, overall, confirm that the leakage of the active should be limited linking it to a polymeric backbone by a covalent bond.

Concerning the morphological results from cryo-TEM investigation, it still remains to be established the structure of the thick layer observed around the *N*,*O*-CMCS-DA NPs. In our opinion, it may be due to physisorption/chemisorption phenomena of the hydrophilic polymer PVA, as well as to a hydrated polymer corona thickness due to high vacuum TEM environment [43].

In the FT-IR spectrum of *N*,*O*-CMCS-DA NPs it is present a clear absorption peak at 1731 cm^−1^ which is absent in the starting *N*,*O*-CMCS-DA. In our opinion, this absorption band at 1731 cm^−1^ may suggest the presence, at least in part, of *N*,*O*-CMCS in the –COOH form which is reported to occur at 1741–1737 cm^−1^ [47]. In other words, it is possible that, under the conditions used to prepare the *N*,*O*-CMCS-DA NPs, some of the carboxymethyl groups of the polymeric chains are dissociated (–COONa) and others are in unionized form (–COOH) [53].

However, solid state studies on *N*,*O*-CMCS-DA NPs were mainly performed in order to gain insights on the internal structure and surface features of these nanocarriers at solid state and to assess their thermal stability Overall, the results from DSC analysis (Figure 2) suggest that the *N*,*O*-CMCS-DA NPs seem characterized by an amorphous internal structure which, at temperatures greater than 260 °C, undergo thermal decomposition. TGA runs (Figure 3) have been performed to determine the thermo-stability of the nanoformulations, as already reported elsewhere [54]. Overall, all TGA observations revealed the higher thermal stability of the nanoparticle formulation for potential application in pharmaceutical preparations. Therefore, it could be hypothesized that the DA decomposition occurred at a temperature of about 322 °C for *N*,*O*-CMCS-DA NPs is slightly slower than that observed on pure DA•HCl and, furthermore, in good agreement with DSC results of thermal stability (Figure 2). However, since the *N*,*O*-CMCS and DA•HCl decompositions fell in a narrow temperature range (i.e., 295–335 °C), a precise evaluation of the DA content by TGA analysis was not possible. XPS analysis evidenced an increase in the amide groups percentage on the *N*,*O*-CMCS-DA NPs surface with respect to CMCS. This finding clearly indicated the presence of amide bonds between the active compound (DA) and the polymer.

As for the stability of *N*,*O*-CMCS-DA NPs on storage, the results reported in Figure 5a–c suggest that, as expected, the best storage conditions occur at the lowest temperature investigated, namely at 4 °C.

In vitro release tests from *N*,*O*-CMCS-DA NPs were carried out in SNF supplemented by 0.25% (*w*/*v*) of mucin and it because such a release medium is more biomimetic than SNF alone previously adopted for release studies starting from pure *N*,*O*-CMCS-DA amide conjugate [30]. In fact, it is well known that nasal epithelium is covered by a mucus gel. Drug molecules or particles in this environment can interact with mucin chains and it markedly influences their transport through the olfactory mucus layer and, hence, the amount that could be transported into brain via the olfactory and trigeminal nerves [6]. In this context, it is recognized that a challenging aspect in the research field of nose-to-brain delivery is to provide evidence, if any, suggesting transport through olfactory mucus of intact nanocarriers [6]. Unfortunately, in the release experiments we cannot use mucin concentrations close to that occurring in the airway (about 3% *w*/*v*) [55] due to the considerable viscosity of the resulting medium that could prevent a correct determination of DA and *N*,*O*-CMCS-DA conjugate by HPLC. Hence, as above mentioned, the cumulative DA released shown on *y*-axis of Figure 5d should be actually intended as pure neurotransmitter or *N*,*O*-CMCS-DA conjugate or a mixture of both. As can be seen from Figure 5d, a prompt release of DA occurs, reaching 20% of DA delivery after 3 h and this amount resulted constant up to more than two days of release. We hypothesized that this prompt delivery corresponds to the diffusion of a mixture of low molecular weight fractions of *N*,*O*-CMCS-DA conjugate from swollen *N*,*O*-CMCS-DA NPs with core-corona structure and pure neurotransmitter produced by hydrolytic cleavage of the amide bond on the surface of swollen *N*,*O*-CMCS-DA NPs. In fact, diffusion of water-soluble low molecular weight fractions of *N*,*O*-CMCS-DA conjugate and hydrolytic cleavage of the amide bond may occur in swollen *N*,*O*-CMCS-DA NPs being the polymeric chains more exposed to the release medium. It should be noted that the resulting amount of neurotransmitter could be suitable for brain delivery considering that DA, as neurotransmitter, is a potent biologically active substance. On the other hand, it should be also considered that the in vivo release conditions can be markedly different from those selected in vitro due to possible presence of amidase enzymes which catalyse the hydrolytic cleavage of the amide bond increasing so the amount of neurotransmitter released. Hence, in vivo DA release could be triggered in a greater amount.

Since *N*,*O*-CMCS-DA NPs were prepared from the *N*,*O*-CMCS-DA conjugate which can be classified as a mucoadhesive polymer as previously demonstrated [30] and similar to how it occurs for other chitosan and its derivatives [56,57], it may be reasonably assumed as further feature of *N*,*O*-CMCS-DA NPs that even these nanocarriers should possess good mucoadhesive properties.

To assess the potential use of *N*,*O*-CMCS-DA NPs for nose-to-brain delivery, cytotoxicity tests were carried out on olfactory ensheathing cells (OECs) which are glial cells found in the olfactory system. As shown in Figure 6, NPs were slightly toxic to OECs, being the maximal reduction of cell viability of around 10% at highest DA concentration tested (i.e., 75 μM DA).

However, the most interesting outcome from the biological evaluation is that mucin determined NPs uptake by more cells, as demonstrated by the higher number of cells that had taken up FITC-*N*,*O*-CMCS-DA NPs in the presence of mucin compared to that observed without mucin. To account for this result, it must be considered that NPs used in biological application are exposed to extracellular proteins that can be adsorbed on the surface of these nanocarriers to form a protein corona layer. The adsorbed protein coat influences physicochemical properties of such NPs including release profile [58], their trafficking [59], and their subsequent interactions with cells [60]. Hence, these surface modified NPs with natural protein possess properties very different from chemically surface modified NPs so that the formers can be considered at the interface of biomimetic NPs and biological systems [61]. The protein layer can be further classified as *hard* or *soft* corona depending on whether the coat is tightly or loosely bound to the nanocarrier, respectively [62].

Herein, we evaluated the effect of the protein mucin, which is present in the mucus gel covering the nasal epithelium at about 3% *w*/*v* concentration [55], for in vitro release profile and uptake studies. As for the effect on drug release, it has been suggested that the protein layer on the nanoparticle surface acts as a shield which reduces or impedes at all drug release leading to a sustained release. Instead, as shown in Figure 5d, we noted a prompt release and this behaviour, in our opinion, can be explained by the very low mucin concentration used in these experiments (i.e., 0.25% *w*/*v*). Under such conditions, indeed, we believe that a *soft* corona layer can be formed at most and for such particles, prompt release (burst effect) have been described just because the mentioned shielding effect of the resulting protein corona does not occur [58,63].

In contrast, in the cell uptake experiments, we verified a strong effect of the mucin at physiological concentration significantly influencing the interaction with OECs. In the presence of mucin, the number of cells that had taken up by OECs increased in comparison with that observed in the absence of mucin. At present, it is not simple to account for this outcome but our hypothesis is, firstly, that mucin may be adsorbed onto the surface of *N*,*O*-CMCS-DA NPs forming a *hard* corona structure by hydrogen bonding interactions between the –OH groups of the polymer backbone and the hydroxylated amino acid (i.e., threonine and serine) present in mucins [55,64]. Moreover, even π-π hydrophobic interactions between aromatic amino acids of mucins and benzene ring of DA may be involved [65]. On the other hand, it is well known that OECs surround the olfactory axons from the epithelial membrane up to lamina propria of the olfactory mucosa and different types of OECs are expressed at different localizations (e.g., OECs located in the mucosa, OECs resident in the lamina propria etc.) [8]. OECs can be considered a gateway to the central nervous system, since they line the olfactory nerve during its course from the olfactory mucosa to the olfactory bulb, reaching also the cerebrospinal fluid [5]. We further hypothesized that mucin-*N*,*O*-CMCS-DA NPs complex may bind to specific mucin sites on the OECs located in the epithelial membrane of the olfactory mucosa and this can increase the *N*,*O*-CMCS-DA NPs uptake compared to that observed in the absence of mucin. Indeed, under physiological conditions, mucins fibers are expressed on the membrane surface of the mucosa of epithelial cells and they have a common framework consisting of repeated PTS (proline-threonine-serine) backbone with intermitting cysteine rich domains [8] (Figure 9). Binding of *N*,*O*-CMCS-DA NPs to mucin fibers may occur via thiol-disulfide exchange between thiol containing amino-acids (i.e., cysteine) of cell surface mucin fibers and disulfide bonds of mucin proteins adsorbed on the surface of NPs. Once anchored to the cell surface, *N*,*O*-CMCS-DA NPs uptake by OECs may take place via endocytosis mechanisms. To support this hypothesis, it has been found that mucin corona enhanced the cell internalization of polystyrene-benzopyrene NPs by lung epithelial cells (A549 cell line) through micropinocytosis [65]. It has been also observed that the mucin corona coating and resultant nano-clusters of gold nanoparticles dramatically facilitated the endocytosis of nanoparticles by intestinal epithelial cells by favouring the membrane interaction of NPs [66].

Finally, the limits of our work should be considered. Our simplistic model does not take into account the many obstacles that NPs have to face for an efficient nose-to-brain delivery of DA. Mucus and mucociliary clearance are the first barrier that NPs encounter to limit their arrival on the olfactory mucosa. When NPs reach the olfactory epithelium, they have to traverse the nasal olfactory epithelium and, depending on the pathway followed, also the arachnoid membrane surrounding the subarachnoid space [67,68]. As concerning the mucociliary clearance, the mucoadhesiveness of *N*,*O*-CMCS-DA NPs renders unlikely this mechanism as opposing to the NP delivery of DA. Moreover, the internalization of *N*,*O*-CMCS-DA NPs by OECs was already considerable at 2 h of incubation. The mucus flow rate in the nose is 5 mm/min (with a range of 0.5–23.6 mm/min) and hence the mucus layer is renewed every 15–20 min [67,69]. Considering this parameter, it may be possible that mucoadhesive properties of *N*,*O*-CMCS-DA NPs permit the increase of residence time and internalization by olfactory nerve terminations [8,69]. Another variable that our study does not consider is the breathing pattern. Since the ultimate method to delivering NPs for nose-to-brain will be most likely through spray devices, the nasal deposition following spraying has been studied by different variables, including airflow rate. The association between the inspiratory airflow rate and aerosol deposition patterns in the nose has been considered for aerosol deposition in the olfactory region for nose-to-brain delivery, although there is much debate about it, with contrasting results depending on the used model [70]. Thus, all these parameters should be considered in the evaluation of possible resistances that NPs can meet in the process of nose-to-brain delivery of their payload.

## 5. Conclusions

Herein, we have successfully prepared *N*,*O*-CMCS-DA NPs by nanoprecipitation method starting from the *N*,*O*-CMCS-DA amide conjugate where DA was covalently linked on the polymeric backbone. The obtained nanocarriers were characterized by PCS analysis which showed a mean diameter value of 289 ± 50 nm and a bimodal size distribution. Such particle features should be due to intrinsic factors of the preparative method as well as to the high polydispersion in molecular weight (30–500 kDa) of the starting polymer used (*N*,*O*-CMCS). It appears that a notable improvement of the physicochemical properties of *N*,*O*-CMCS-DA NPs should result optimizing both the reaction conditions and removing the excess of surfactant (PVA) used. The zeta potential values observed (−32.4 ± 1.6 mV) suggest that these particles should possess sufficient colloidal stability. Moreover, the high percentages of E.E.% both for the neurotransmitter (DA) and for the fluorescent probe (FITC) confirm that the leakage of the cargo should be limited linking it to a polymeric backbone by a covalent bond. Solid state studies through FT-IR, DSC and thermogravimetric analysis showed that the particles were amorphous and demonstrated their thermal stability for potential application in pharmaceutical preparations. XPS analysis supplied evidence of the amide bond between DA and CMCS and the absence of free-DA on the NPs surface. Cell-viability tests on OECs suggested only a slight cytotoxicity at the highest DA concentration used (75 μM). Finally, it was shown by epifluorescence microscopy that mucin determined *N*,*O*-CMCS-DA NPs uptake by more cells. This result can be accounted for considering that mucin may be adsorbed onto the surface of *N*,*O*-CMCS-DA NPs leading to a *hard* corona structure. This mucin- *N*,*O*-CMCS-DA NPs complex may bind to specific cell surface mucin sites and it can favour the *N*,*O*-CMCS-DA NPs uptake.

Altogether, these findings seem promising for further development of *N*,*O*-CMCS-DA NPs for nose-to-brain delivery application in PD.

## Figures and Tables

**Figure 1 pharmaceutics-14-00147-f001:**
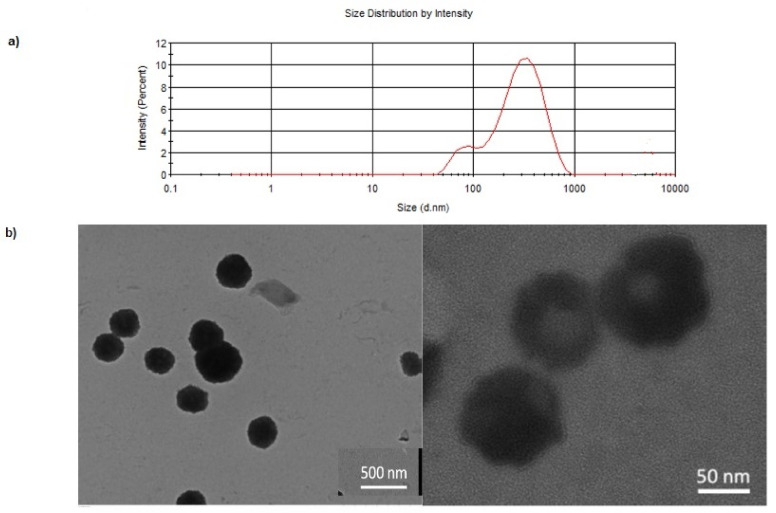
(**a**) Particle size distribution and (**b**) Cryo-TEM images of *N*,*O*-CMCS-DA NPs.

**Figure 2 pharmaceutics-14-00147-f002:**
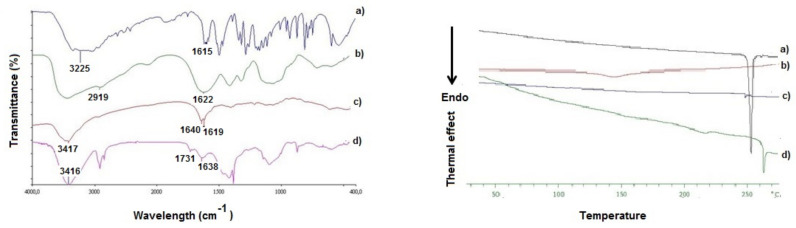
FT-IR spectra (left panel) and DSC profiles (right panel) of (**a**) pure DA•HCl; (**b**) *N*,*O*-CMCS; (**c**) *N*,*O*-CMCS-DA; (**d**) lyophilized *N*,*O*-CMCS-DA NPs.

**Figure 3 pharmaceutics-14-00147-f003:**
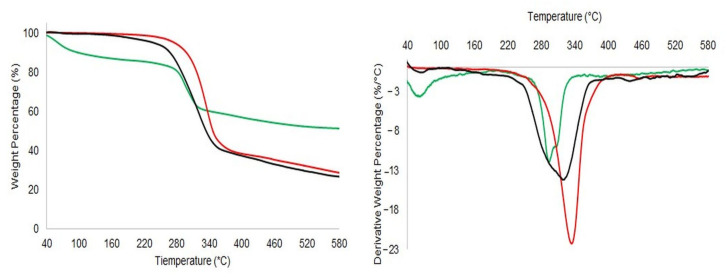
TG (on the left) and DTG (on the right) curves of pure DA (red line) *N*,*O*-CMCS (green line) and *N*,*O*-CMCS-DA NPs (black line).

**Figure 4 pharmaceutics-14-00147-f004:**
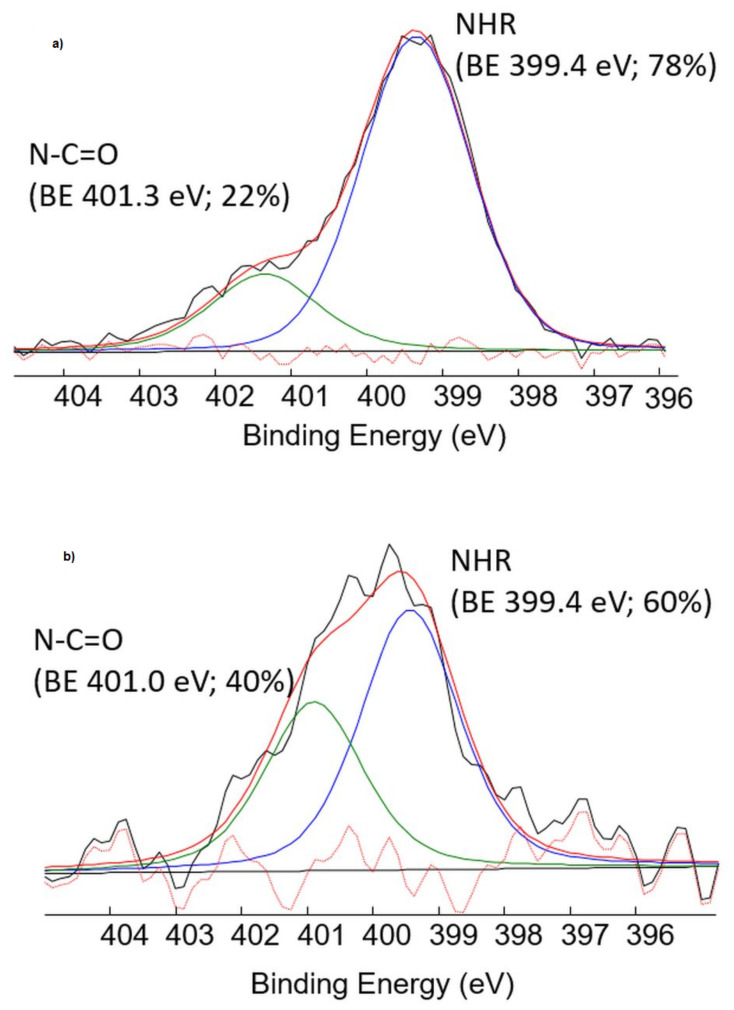
N1s curve fittings of (**a**) *N*,*O*-CMCS and (**b**) *N*,*O*-CMCS-DA NPs samples. Uncertainty on BE peak positions was ±0.2 eV.

**Figure 5 pharmaceutics-14-00147-f005:**
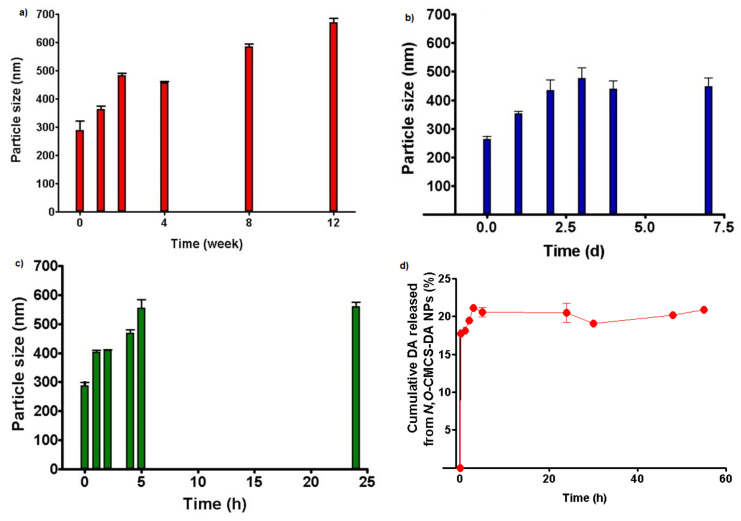
Particle size variation of *N*,*O*-CMCS-DA NPs after incubation at: (**a**) 4 °C; (**b**) 25 °C; (**c**) 37 °C. Panel (**d**): DA (or *N*,*O*-CMCS-DA or a mixture of both, see text) released from *N*,*O*-CMCS-DA NPs in Simulated Nasal Fluid (SNF) supplemented with 0.25% (*w*/*v*) of mucin.

**Figure 6 pharmaceutics-14-00147-f006:**
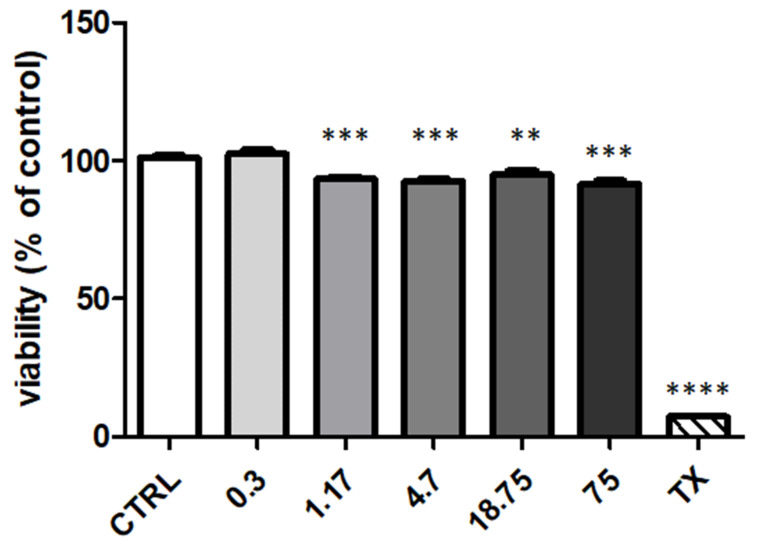
Cytotoxicity of *N*,*O*-CMCS-DA NPs. OECs were challenged with *N*,*O*-CMCS-DA NPs at the indicated DA concentrations of 0.3, 1.17, 4.7, 18.75, and 75 μM. Cells were then assayed for vitality by the MTT assay. Controls (CTRL) are untreated cells (100% of vitality), whereas 1% Triton X-100 (TX) was used as positive control. ** *p* < 0.01, *** *p* < 0.001 vs. CTRL; **** *p* < 0.0001 vs. TX. Data are the results of two experiments each carried out in six wells.

**Figure 7 pharmaceutics-14-00147-f007:**
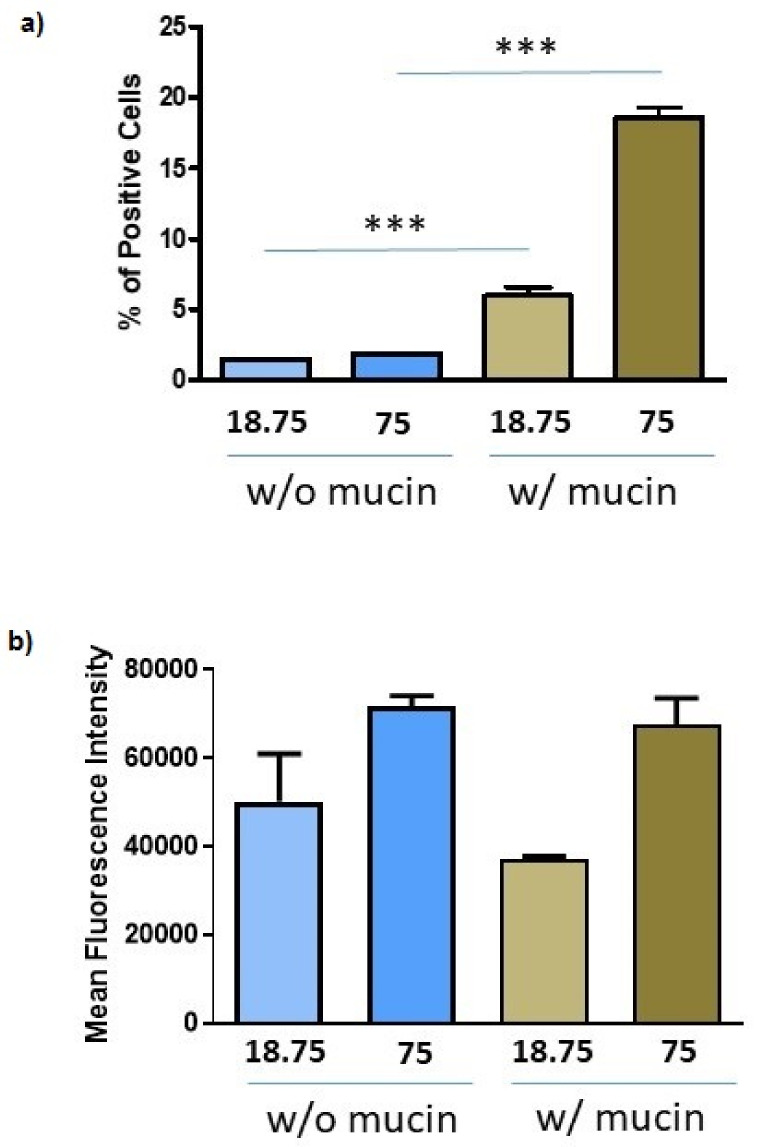
Cellular uptake of FITC-*N*,*O*-CMCS-DA NPs by OECs. OECs were incubated with FITC-*N*,*O*-CMCS-DA NPs at DA concentrations of 18.75 and 75 μM in the presence or absence of 2.5% mucin for 2 h and evaluated by flow cytometry. Positive cells, shown as percentages (**a**), and the mean fluorescence intensity (**b**), were obtained in two experiments each conducted in duplicate and shown as mean ± SD. In (**a**) *** *p* < 0.001 for both DA concentrations, w/o mucin vs. w/mucin.

**Figure 8 pharmaceutics-14-00147-f008:**
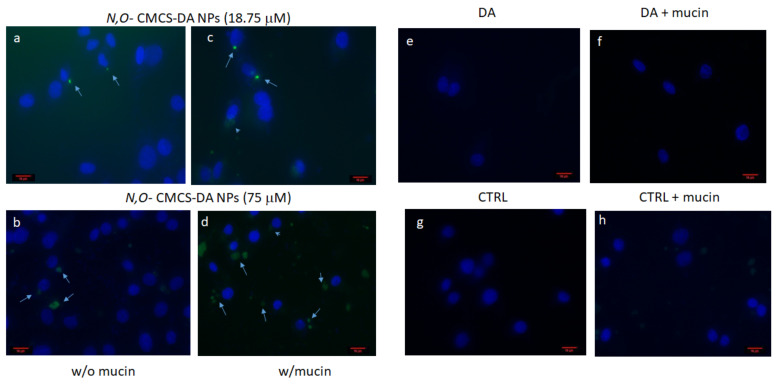
Epifluorescence microscopy of OECs incubated with FITC-*N*,*O*-CMCS-DA NPs at DA concentrations of 18.75 (**a**,**c**) and 75 μM (**b**,**d**), or FITC-*N*,*O*-CMCS-DA 75 μM (**e**,**f**), were incubated with OECs in the presence or absence of mucin for 2 h and then evaluated by epifluorescence microscopy. Controls (CTRL) were cells incubated with medium only in the presence or absence of mucin (**g**,**h**). Arrows indicate NPs in close vicinity of nuclei as dots, while arrowheads point to more diffuse perinuclear staining. Bar = 10 μm.

**Figure 9 pharmaceutics-14-00147-f009:**
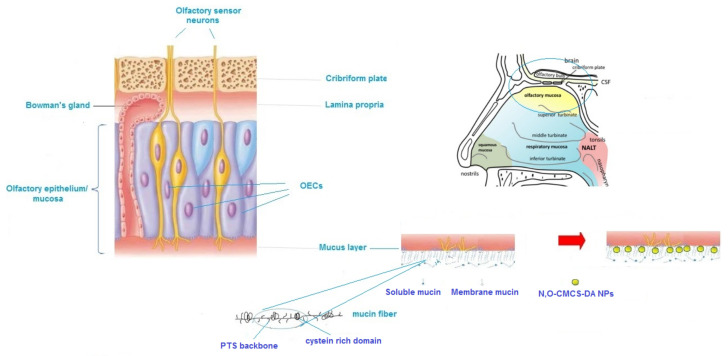
Possible mechanism accounting for the enhanced uptake of *N*,*O*-CMCS-DA NPs by OECs in the presence of mucin.

**Table 1 pharmaceutics-14-00147-t001:** Physicochemical characterization of NPs prepared ^a^.

Formulation	Size (nm)	PDI ^a^	Zeta Potential (mV)	E.E. DA (%)	E.E. FITC (%)
*N*,*O*-CMCS NPs	608 ± 58	0.47–0.62	−9.2 ± 0.7	-	-
*N*,*O*-CMCS-DA NPs	289 ± 50	0.48–0.54	−32.4 ± 1.6	94 ± 3	-
FITC*-N*,*O*-CMCS NPs	252 ± 33	0.50–0.59	−20.3 ± 1.0	-	99.9 ± 0.0
FITC*-N*,*O*-CMCS-DA NPs	425 ± 28	0.34–0.36	−14.2 ± 1.6	89 ± 2	97.4 ± 0.7

^a^ PDI: polydispersity index.

**Table 2 pharmaceutics-14-00147-t002:** Surface composition of DA, CMCS and *N*,*O*-CMCS-DA NPs samples.

Sample	Atomic Percentage %					
	C1s	O1s	N1s	Cl2p	Na1s	Si2p
pure DA•HCl	71.7	15.3	6.8	6.1	--	--
CMCS	51.5	32.3	8.8	3.8	3.6	--
*N*,*O*-CMCS-DA NPs	74.2	22.2	0.6	0.8	0.7	1.5

## Data Availability

Not applicable.

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
