# Peer review of "Novel Nanoparticles Based on N,O-Carboxymethyl Chitosan-Dopamine Amide Conjugate for Nose-to-Brain Delivery"

_pharmaceutics, 2022, doi:10.3390/pharmaceutics14010147_

Round 1
Reviewer 1 Report
In this manuscript, the authors studied the N,O-CMCS-DA NPs characteristics, i.e., internal structure, thermal stability, surface features, release profile and physical stability, via. nose-to-brain delivery. It is interesting, while some comments are needed to be addressed before further acceptance consideration., i.e.,
- What does DA represent in Abstract and in the paper? Neurotransmitter dopamine (Line 85) or Dopamine hydrochloride (Line 137)?
- Have the authors considered the mucociliary clearance effects (see Fig.9) which can prevent NPs entering brain via. nose? Please provide quantitative estimation.
- Could the breathing patterns affect the delivery efficiency of NPs via. nose-to-brain? Since it will affect the mucus/mucin movements (i.e., NPs in the mucus/mucin) in the olfactory.
- Continuing with question 3, have the authors considered the possible resistances that the NPs can meet in the process?
- Why did the authors employ porcine stomach mucin rather than human olfactory mucus? Have the authors compared the different properties of these two mucins?
- Grammar and typo issues should be addressed throughout the manuscript.
Author Response
In this manuscript, the authors studied the N,O-CMCS-DA NPs characteristics, i.e., internal structure, thermal stability, surface features, release profile and physical stability, via. nose-to-brain delivery. It is interesting, while some comments are needed to be addressed before further acceptance consideration
We thank the Reviewer for his/her overall positive opinion about our work and for giving us the opportunity to improve the quality of presentation of the manuscript.
Q1. What does DA represent in Abstract and in the paper? Neurotransmitter dopamine (Line 85) or Dopamine hydrochloride (Line 137)?
R1. In the revised version of the manuscript, we make it clear that "DA" refers to "neurotransmitter dopamine" while "DA•HCl" refers to "Dopamine hydrochloride".
Q2. Have the authors considered the mucociliary clearance effects (see Fig.9) which can prevent NPs entering brain via. nose? Please provide quantitative estimation.
R2. It is well known that nasal mucociliary clearance contributes to the body’s defense mechanisms by entrapping potentially dangerous substances that may be rapidly eliminated. However, as remembered by the Reviewer, the mucociliary clearance can also prevent NPs entering brain via nose. We took into account this possible limiting factor in nasal drug delivery. However, as already detailed in the Discussion, N,O-CMCS-DA NPs were prepared from the N,O-CMCS-DA conjugate which can be classified as a mucoadhesive polymer as previously demonstrated (Di Gioia et al., 2021, doi: 10.1016/j.ijpharm.2021.120453) and just like occurs for other chitosan and its derivatives (Palazzo et al., 2017, doi: 10.1016/j.ejpb.2017.04.020; Trapani et al., 2014, doi: 10.1021/bm401733p). Moreover, the internalization by OECs was already considerable at 2 h of incubation. The mucus flow rate in the nose is 5 mm/min (with a range of 0.5- 23.6 mm/min) and, hence, the mucus layer is renewed every 15 -20 min (Pardeshi and Belgamwar, 2013, doi: 10.1517/17425247.2013.790887). Thus, considering that the t1/2 mucociliary clearance in the nasal cavity is 20 min in humans (Sonvico et al., 2018, doi: 10.3390/pharmaceutics10010034), it may be possible that mucoadhesive properties of N,O-CMCS-DA NPs allow to increase residence time and internalization by different pathways, including olfactory nerve terminations (Ganger and Schindowski, 2018, doi: 10.3390/pharmaceutics10030116; Sonvico et al., 2018).
We have added this quantitative estimation in the last paragraph of the Discussion (Lines 692-697).
Q3. Could the breathing patterns affect the delivery efficiency of NPs via. nose-to-brain? Since it will affect the mucus/mucin movements (i.e., NPs in the mucus/mucin) in the olfactory.
R3. Since the ultimate method to delivering NPs via nose-to-brain will be most likely through spray devices, the nasal deposition following spraying has been studied by different variables, including airflow rate. The association between the inspiratory airflow rate and aerosol deposition patterns in the nose has been considered for aerosol deposition in the olfactory region for nose-to-brain delivery, although there is much debate about it (Maaz et al., 2021; doi: 10.3390/ pharmaceutics13071079). While some studies have concluded that there is no or only scant effect of the breathing profile on deposition (Warnken et al, 2018, doi: 10.1021/acs.molpharmaceut.7b00702; Guo et al., 2005; doi: 10.1007/s11095-005-7391; Foo et al., 2007, doi: 10.1089/jam.2007.0638), others showed deeper deposition beyond the nasal valve by increasing the flow rate (Nižić et al., 2019, doi: 10.1016/j.ijpharm.2019.04.015; Xu et al., 2020, doi: 10.1021/acs.molpharmaceut.0c00372; Moraga-Espinoza et al., 2018, doi: 10.1016/j.ijpharm.2018.06.058). These data have been obtained with nasal replicas but in non-physiological experimental conditions, e.g., steady airflow rather than tidal inhalation patterns, lack of mucus, and no consideration of cilia (Maaz et al., 2021), thereby more physiological models are warranted to consider the airway flow rate and hence breathing patterns in the delivery efficiency of NPs in this context.
We have added some of these issues in the last paragraph of the Discussion (Lines 687-707).
Q4. Continuing with question 3, have the authors considered the possible resistances that the NPs can meet in the process?
R4. It is well known that the sticky and viscous nasal mucus combined with mucociliary clearance oppose a resistance that NPs can meet in the process of delivering DA to the brain (Ganger and Schindowski, 2018). In this regard, some strategies were developed to control the interaction of nanocarriers with the mucus in order to achieve effective formulations able to penetrate the nasal mucus reaching the underlying epithelium and to minimize mucocilliary clearance. The main strategies which facilitate mucopenetration of NPs have been recently reviewed (Schattling P. et al, Macromol. Biosci. 2017, 17, 1700060) focusing essentially three approaches denoted as, respectively, low molecular weight PEG (and other hydrophilic polymers) coating, use of zeta-potential shifting polymers and virus-like particles. In particular, it has been noted that PVA coated NPs show high mucoadhesion but reduced mucopenetration (J. Control. Release 192 (2014) 202–208). Hence, the need to optimize N,O-CMCS-DA NPs preparative method and removal the excess of surfactant pointed out in Discussion and Conclusions Sections of the manuscript is further supported by the possibility to achieve effective mucopenetration by the particles.
Overall, we have added a novel paragraph at the end of the Discussion taking into account all these various issues (Lines 687-707).
Q5. Why did the authors employ porcine stomach mucin rather than human olfactory mucus? Have the authors compared the different properties of these two mucins?
R5. We used porcine stomach mucin instead of human olfactory mucus since the former is commercially available, while the second is not. Actually, there are not specific studies in which the properties of the human olfactory mucus have been compared with those of porcine stomach mucin. Nevertheless, the healthy mucus made with porcine mucin and buffer salts diluted in a buffer solution reflects that of healthy individuals (Wang et al., 2016, doi: 10.1016/j.jpha.2016.05.003). Pig mucus and human mucus are similar in structure and molecular weight (Kararli et al., 1995, doi: 10.1002/bdd.2510160502), which is important given that it is also possible to observe differences in mucus properties for different animals of the same species (Groo et al., 2013, doi: 10.2147/IJN.S51837).
Morover, in the airways, MUC5AC and MUC5B are the major polymeric mucins secreted from goblet cells in the surface epithelium. MUC2 is present in the respiratory tract, albeit in lower concentrations; in the stomach, the major secreted mucin present is MUC2, followed by MUC5AC, and lower concentrations of MUC5B, MUC6, and MUC7 (Leal et al., 2017, doi: 10.1016/j.ijpharm.2017.09.018)]. Hence, most mucin types occur in both the organs considered, although in different amounts.
In our opinion, based on these features as well as considering that mucins share a high homology degree (Ganger and Schindowski, 2018), the results obtained by using porcine stomach mucin can be considered significant, even though human olfactory mucus was not used.
Mucins and expression in airways and stomacha
Organs |
Secreted, oligomeric gel forming mucins |
Secreted, monomeric non gel forming mucins |
Cell surface associated mucins |
Airways |
MUC2, MUC5AC, MUC5B, MUC19 |
MUC7 |
MUC1 ,MUC16, MUC20 |
Stomach |
MUC5AC,MUC6 |
|
MUC1, MUC12, MUC13, MUC17 |
aFrom Leal et al., Int. J. Pharm. 532 (2017) 555–572.
Q6. Grammar and typo issues should be addressed throughout the manuscript.
R6. An in-depth check of the issues evidenced by the Reviewer has been performed and corrected in the revised form.

Reviewer 2 Report
This manuscript by Trapani et al describes encapsulation of dopamine and dopamine amide derivative in PVA nanoparticles. They determined particles sizes over time, release study and some some biological evaluation. I think this manuscript can be published after extensive English language editing. Another major concern is, they did not mention if chitosan amide derivative of dopamine has potential to treat the condition in question and also they did not report any specific dopamine release study from this derivative. This is important as dopamine and its chitosan linked amide derivatives might have different pharmacological properties.

Author Response
This manuscript by Trapani et al describes encapsulation of dopamine and dopamine amide derivative in PVA nanoparticles. They determined particles sizes over time, release study and some biological evaluation.
We did not encapsulate neither the neurotransmitter dopamine nor dopamine hydrochloride. Furthermore, PVA was the surfactant adopted by us according to the nanoprecipitation method as reported in Section 2.3. rather than the polymer forming nanostructures. Indeed, nanoparticles were achieved from the conjugate N,O-CMCS-DA amide subjected to the nanoprecipitation method.
Q1. I think this manuscript can be published after extensive English language editing.
R1. An in-depth check of the English language has been performed and, in this regard, the text has been corrected where necessary. We thank the Reviewer for his/her overall positive opinion about our work.
Q2. Another major concern is, they did not mention if chitosan amide derivative of dopamine has potential to treat the condition in question and also they did not report any specific dopamine release study from this derivative. This is important as dopamine and its chitosan linked amide derivatives might have different pharmacological properties.
R2. The potential of the chitosan amide derivative of dopamine (i.e., N,O-CMCS-DA) for the treatment of PD as well as the neurotransmitter release profile from this derivatives has already been described by us in a previous paper i.e., Ref. [30] of the manuscript. In the revised form, we have added a sentence evidencing that the potential of N,O-CMCS-DA amide conjugate to treat PD has been previously described together with its release profile in Ref [30] (see Lines 494-495).
Moreover, as stated in Section 3.6 of the submitted manuscript (as well as in our previous work Int J Pharm 2021, 599, 120453), for uptake studies we have selected DA concentrations from the pure N,O-CMCS-DA amide conjugate and from N,O-CMCS-DA amide conjugate NPs in the range 18.75–75 μM (corresponding to 2.9–11.5 μg/mL) because, according to Tang et al.’s work (Drug Deliv 2019, 26, 700-707) to alleviate motor symptoms in a rat model of PD the dose of ~ 7.4 μg/mL of DA is mandatory for intranasal administration.
On the base of these in vitro results, currently, work is in progress for the comparison of the pharmacological effects of N,O-CMCS-DA ester conjugate, N,O-CMCS-DA amide conjugate and nanoparticles based on N,O-CMCS-DA amide conjugate in order to identify the most promising drug delivery system for intranasal administration of DA to parkinsonian patients.
Concerning the issue arising in the PDF file sent by this Reviewer “From figure 1a, the size of this particles is almost 500 nm”, in Figure 1a there is a cluster of ten particles where, by using the scale bar, nine particle diameters measured less than 500 nm and only one particle showed size about 500 nm. In our opinion, mean diameters from TEM pictures could be in agreement with particle size from PCS (289 ± 50 nm).

Round 2
Reviewer 2 Report
The authors have satisfactorily attended to my concerns